# An Oligopeptide-Protected Ultrasmall Gold Nanocluster with Peroxidase-Mimicking and Cellular-Imaging Capacities

**DOI:** 10.3390/molecules28010070

**Published:** 2022-12-21

**Authors:** Daoqing Fan, Jiale Ou, Ling Chen, Lichao Zhang, Zhiren Zheng, Haizhu Yu, Xiangming Meng, Manzhou Zhu

**Affiliations:** 1Department of Chemistry and Centre for Atomic Engineering of Advanced Materials, Key Laboratory of Structure and Functional Regulation of Hybrid Materials of Physical Science and Information Technology and Anhui Province Key Laboratory of Chemistry for Inorganic/Organic Hybrid Functionalized Materials, Anhui University, Hefei 230601, China; 2Institute of Energy, Hefei Comprehensive National Science Center, Hefei 230601, China

**Keywords:** gold nanocluster, HRP-mimicking activity, water-soluble, cell imaging

## Abstract

Recent decades have witnessed the rapid progress of nanozymes and their high promising applications in catalysis and bioclinics. However, the comprehensive synthetic procedures and harsh synthetic conditions represent significant challenges for nanozymes. In this study, monodisperse, ultrasmall gold clusters with peroxidase-like activity were prepared via a simple and robust one-pot method. The reaction of clusters with H_2_O_2_ and 3,3′,5,5′-tetramethylbenzidine (TMB) followed the Michaelis-Menton kinetics. In addition, in vitro experiments showed that the prepared clusters had good biocompatibility and cell imaging ability, indicating their future potential as multi-functional materials.

## 1. Introduction

In recent years, biomimetic nano-enzymes have attracted increasing research interest [1,2,3] because of their promising catalytic efficiency and capacity to overcome the disadvantages of natural enzymes, such as high separation cost, low stability and storage difficulties. To date, significant progress has been made with metal nanoparticles protected by organic/biological ligands. Gold and its alloy nanoclusters represent one of the most promising categories and have been successfully developed to exhibit peroxidase-like, catalase-like and oxidase-like activities, demonstrating great potential for biosensing and bio clinic applications [4,5]. In this context, horseradish peroxidase (HRP) mimetic represents one of the most attractive subjects because of the diversity of its applications. For example, Li and co-workers recently developed a peroxidase-like nanozyme by organizing gold nanoparticles with single stranded DNA scaffolds and regulating the catalytic activity by changing the scaffold sequence [6]. In addition, peroxidase-like ultrafine gold aerogels have been developed by Shang and co-workers, using D-penicillamine-stabilized gold nanoclusters as the building blocks [7].

Despite the great progress made and highly promising applications, the synthetic procedures for obtaining the current horseradish peroxidase mimetic nano-enzymes generally require relatively harsh conditions. For example, a frequently used strategy for the formation of hybrid materials is to coat the active metal nanoclusters with carriers [8,9,10]. And the one-pot synthesis method always requires strong alkaline conditions, high temperatures (100–200 °C), comprehensive synthetic and separation procedures, or a long reaction time [11,12,13,14,15,16,17]. Consequently, the development of a peroxidase-like nanozyme with easily achievable and mild synthetic and separation procedures is of significant interest.

Compared to plasmonic metal nanoparticles or clusters protected by biological ligands (such as BSA), ultrasmall gold nanoclusters with a sub nanometer (1–2 nm) size-regime have recently emerged as a novel material because of their excellent physicochemical properties (such as ease of functionalization, strong luminescence, and enhanced permeability and retention effects) [18,19]. Specifically, the combination of a cysteine-containing peptide fragment with another biofunctional fragment has been successfully developed as an efficient strategy in the ligand design of metal nanoclusters. In this study, using the DGECGC oligopeptide (note: the DGEA fragment is able to target the overexpressed α_2_β_1_-integrins on the surface of human prostate cancer cells [20,21,22], and the thiol group in the GC fragment helps in size control), a monodispersed, ultrasmall gold nanocluster (~1.3 nm) was prepared via a convenient one-pot synthesis under mild conditions (in water solution, 70 °C, 2 h). The synthesis eliminated the necessity for pre-functionalization or postseparation procedures, as shown in Figure 1. The prepared clusters showed high stability and were brightly emissive after being kept in the dark at 4 °C for over one month. The peroxidase mimetic character was evidenced by the reaction of the as-prepared clusters with H_2_O_2_ and 3.3′,5.5′-tetramethylbenzidine (TMB). Additionally, cytotoxicity tests showed the high biocompatibility and cellular-imaging capacity of the prepared clusters. Therefore, these clusters can be used as an easily achievable, low-toxicity and highly sensitive fluorescence imaging material for future studies.

## 2. Results and Discussion

### 2.1. Characterization of Materials

A reported synthetic method was used to protect the gold nanoclusters with DGEAGC oligopeptide [23]. Briefly, the oligopeptide was mixed with HAuCl_4_ in ultrapure water, followed by thermal reduction at 70 °C. The color of the solution changed from yellow to colorless within 5 min. A pale yellow solution was obtained after ~2 h, indicating the formation of peptide-protected gold nanoclusters (abbreviated as PGN). As shown in Figure 1a, the absorption of the PGN exhibited a shoulder peak at ~400 nm. The absence of a surface plasmon resonance peak at ~520 nm excludes the formation of Au NPs [24,25]. Strong orange-red fluorescence was observed under 365 nm UV light irradiation (Figure 1a inset). The fluorescence spectra showed maximum excitation and emission peaks at 400 nm and 600 nm, respectively. The consistency of optical absorption and the excitation peak indicated that the fluorescence originated from intrinsic transitions (rather than aggregation induced emission, etc.) (Appendix A) [26,27]. After purifying the prepared nanoclusters using native polyacrylamide gel electrophoresis (PAGE), only one band with orange-red luminescence was observed (under UV-light irradiation, (Appendix A). The optical absorption and emission of the fluorescent band components were close to that of the prepared clusters (Appendix A), evidencing their predominance in the crude products. 

Furthermore, HRTEM revealed the uniform size of the clusters with an average size of 1.30 ± 0.01 nm via the Gaussian fitting curve (Figure 1b). In this context, both the optical tests and the TEM analysis indicated the monodispersity of the prepared clusters. In addition, the absolute quantum yields of the PGN were 9.77% and 0.52% in the solid state and aqueous phases, respectively. The average fluorescence lifetime was 6.14 µs (Appendix A) [28,29,30]. According to X-ray photoelectron spectroscopy analysis (XPS, Figure 1c), the binding energies of Au 4f_5/2_ and Au 4f_7/2_ in the PGN were 88.0 and 84.3 eV respectively, corresponding to an Au(I): Au(0) ratio of 67.2%, which indicated the presence of a metallic core structure. The presence of O, C, N, Au, and S elements in the PGN was also verified using XPS (Appendix A) [31,32].

The FTIR spectrum of the PGN was similar to that of free peptide ligands (Figure 1d). The preserved carbonyl peak at 1590 cm^−1^, amide I peak at 1660 cm^−1^, and the carboxylic O-H peaks in the range of 2930–3550 cm^−1^ implied the maintained structure of the peptide ligands on the cluster surface [33,34].

Interestingly, the fluorescence intensity of the PGN was dependent on the temperature and pH of the aqueous solution. As shown in Appendix A, the PGN fluorescence intensity showed a negative linear correlation with temperatures ranging from 20 to 40 °C. With regarding to pH dependency, the solution fluorescence was the strongest at pH = 5 (consistent with the isoelectric point of the C-terminus cysteine amino acid in the oligopeptide ligand).

### 2.2. Peroxidase-like Activity of PGN

As shown in Figure 2a, mixing the aqueous solution of PGN with H_2_O_2_ and TMB resulted in remarkably changed UV-Vis spectra [35], and an additional absorption peak at ~652 nm (characteristic peak of ox-TMB). The observation demonstrated the horseradish peroxidase (HRP)-mimicking activity of the prepared clusters. By contrast, the characteristic peak of ox-TMB was invisible when the free oligopeptide ligand was mixed with H_2_O_2_ and TMB under otherwise identical conditions. Similarly, removing either TMB or H_2_O_2_ from the three-component system resulted in the absence of the ~652 nm peak. The results indicated that the gold nanoclusters, rather than the free ligands, were responsible for the HRP-mimicking activity [36]. It was notable that in addition to H_2_O_2_, the prepared PGN was able to oxidize TMB in the presence of dioxygen, but the significantly lower intensity of the ~652 nm peak indicated the inferior activity of dioxygen compared to that of H_2_O_2_ (Appendix A).

Interestingly, the fluorescence of the PGN was quenched in the three-component system. As shown in Figure 2b, the fluorescence of the PGN did not change after the addition H_2_O_2_ but decreased significantly after TMB was added. Further studies showed that the fluorescence intensity of the PGN-TMB system decreased regularly with the addition of increased amounts of H_2_O_2_ (Appendix A). These phenomena were mainly caused by fluorescence resonance energy transfer (FRET) between the oxidized form of TMB (with optical absorption at 652 nm) and the PGN (with maximum emission at 600 nm, Appendix A) [37,38]. Adding GSH into the PGN + H_2_O_2_ + TMB system, the characteristic absorption of ox-TMB at 652 nm was significantly diminished (Appendix A) and the fluorescence intensity was restored and comparable to that of the PGN + H_2_O_2_ + TMB (Appendix A). The results verified the reaction of GSH with ox-TMB. In view of the GSH induced emission enhancement of PGN, we were not able to completely exclude the possibility that the GSH binds with PGN to restore the fluorescence.

Like other Au NC-based peroxidase mimetics and the natural horseradish peroxidase, the peroxidase activity of PGN was dependent on the solution characteristics, such as the PGN concentration, pH, and temperature [39,40]. As shown in Figure 3a, the optical absorption of the reaction system at 652 nm increased with cluster concentration, and each system showed a linear correlation within 60 min. In addition, the catalytic activity of the PGN was sensitive to the pH and temperature of the solution, similar to recently reported nanomaterial-based peroxidases [41]. Based on the results in Figure 3b,c, pH = 3 and temperature 37 °C were selected for the subsequent tests.

Regarding the catalytic mechanism, it has frequently been suggested that the peroxidase-mimic nanozymes undergo a two-step mechanism, i.e., decomposition of H_2_O_2_ to generate hydroxyl radicals (•OH), and then oxidation of the colorless TMB to blue ox-TMB. To examine whether our PGN followed the same mechanism, we used terephthalic acid (TA) as a fluorescent probe for •OH, because the formed 2-hydroxyterephthalic acid (if present) exhibits an intense, characteristic emission at ~430 nm [42,43]. As shown in Figure 3d, the fluorescence intensity of the aqueous solution of the PGN-TA-H_2_O_2_ system at ~430 nm was stronger than that of the other controls. The result indicated that PGN can generate •OH from H_2_O_2_, thereby promoting the oxidation of TMB.

### 2.3. Kinetic Assay of the Peroxidase-like Activity

Next, we evaluated the kinetics of the peroxidase-like activity of PGN using a typical Michaelis–Menten approach. As shown in Figure 4, in a certain range of H_2_O_2_ and TMB concentrations, a typical absorption intensity spectrum was obtained under the reaction conditions of pH = 3 and 37 °C. The typical curves were obtained with both TMB and H_2_O_2_ as the substrate by monitoring their absorbance change at 652 nm (Figure 5a,c). Then, the enzyme kinetic parameters, Michaelis–Menten constant (K_m_) and maximal reaction velocity (V_max_), could be calculated from Lineweaver–Burk plots (Figure 5b,d). The K_m_ value is a measure of the binding affinity between enzymes and substrates: the higher the value of K_m_, the weaker the affinity. The K_cat_ value measures the maximum number of colored products generated per enzyme per second [44]. As summarized in Table 1, when using TMB as the substrate, the K_m_ and K_cat_ values of the PGN were 0.31 mM and 8 × 10^−5^ s^−1^, respectively. When using H_2_O_2_ as the substrate, the K_m_ and K_cat_ values of the PGN were 1069 mM and 5.35 × 10^−4^ s^−1^, respectively. The K_m_ value of the PGN with TMB as the substrate was almost identical to that of HRP, indicating that the affinity of PGN to TMB is like that of HRP. Additionally, the K_m_ value of PGN for H_2_O_2_ appeared much higher than that of HRP. The results indicate that the affinity of PGN to H_2_O_2_ is lower than that of HRP. Therefore, a higher concentration of H_2_O_2_ is required to obtain the maximum reaction rate of the PGN. Then, we tested the potential peroxidase-mimicking activity of the GSH protected gold nanoclusters. According to the detailed kinetic analysis (Appendix A), the V_max_, K_m_ and K_cat_ of the GSH protected clusters closed to that of the oligopeptide protected ones (Appendix A). The result indicated that the incorporation of the bio-functional DGEA fragment did not affect the biomimicking activity of the metal clusters.

### 2.4. Cytotoxicity Study and In Vitro Imaging

To evaluate the biocompatibility of PGN, cell viability assays were carried out via an MTT assay (MTT = 3-(4,5-dimethylthiazol-2-yl)-2,5 diphenyltetrazolium bromide) [45,46], using NIH 3T3 and HeLa cells as the test candidates. As shown in Figure 6a, the presence of 0–100 μM PGN produced slight perturbation in the proliferation of the NIH 3T3 and HeLa cells within 24 h, and 95% cell viability was maintained even up to a relatively high dose of PGN of 100 μM after 24 h incubation. The low cytotoxicity to NIH 3T3 and HeLa cells is fundamentally important for cell imaging applications. As shown in Figure 7, after 2 h incubation with the PGN, the NIH 3T3 and HeLa cells showed a bright fluorescent signal in the cytoplasm (using confocal laser scanning microscopy), indicating that PGN have cell uptake and biological imaging capabilities.

## 3. Materials and Methods

### 3.1. Materials and Chemicals

All chemicals were commercially available and used without further purification. Custom-made DGEAGC polypeptides were obtained from Bankpeptide Biological Technology Co., Ltd, (Hefei, China). Hydrogen tetrachloroaurate tetrahydrate (HAuCl_4_·4H_2_O, ≥99.99%, metals basis) was purchased from Sino-Platinum Metals Co., Ltd. (Shanghai, China). Hydrochloric acid (HCl, ≥98%); and sodium hydroxide (NaOH, ≥97%) were purchased from Sinopharm Group Co., Ltd, (Beijing, China). Hydrogen peroxide solution(H_2_O_2_, ≥30%) was purchased from Xilong Science Co., Ltd, (Shanghai, China). The 3,3’,5,5’-tetramethylbenzidine (TMB, ≥99%) was purchased from Shanghai Macklin Biochemical Co., Ltd, (Shanghai, China). Terephthalic acid (TA, ≥98%) was purchased from Shanghai Macklin Biochemical Co., Ltd. All glassware was thoroughly cleaned with aqua regia (HCl/HNO_3_ 3/1 *v*/*v*), rinsed with copious amounts of pure water, and then dried in an oven prior to use.

### 3.2. Equipment

UV−vis absorption spectra were recorded using a UV-6000PC instrument, and the solution samples were prepared using ultrapure water as the solvent. All fluorescence spectra were obtained using a HORIBA FluoroMax-4P fluorescence spectrophotometer. Transmission electron microscopy (TEM) and HRTEM images were obtained from a JEM-F200 microscope. X-ray photoelectron spectroscopy (XPS) measurements were performed on a ESCALAB 250 high-performance electron spectrometer with monochromated Al Kα radiation as the excitation source. The binding energies were corrected by referencing the binding energies of C(1s) arising from the added hydrocarbons as an internal standard. Fourier transform infrared spectroscopy (FT-IR) was recorded using a Bruker Vertex80 + Hyperion2000 apparatus. The gold content in the clusters was measured using Inductively coupled plasma mass spectrometry (ICP-MS) on an Agilent 7800 instrument. The fluorescence quantum yield was measured using an Edinburgh-Steady State/Transient Fluorescence Spectrometer FLS1000. Photoluminescence lifetimes were measured by time-correlated single-photon counting (TCSPC) on a Horiba Fluoro max plus spectrofluorometer with a pulsed light-emitting diode (LED) (400 nm) as the excitation source.

### 3.3. Synthesis of the PGN

The fluorescent gold nanoclusters were synthesized using DGEAGC as the surface ligand. Briefly, fresh aqueous solutions of HAuCl_4_ (0.2 g/mL, 68 μL) and the oligopeptide (50 mg/mL, 1.0 mL) were added to 17.6 mL of deionized water. Next, the aqueous solution was vigorously stirred at room temperature for 5 min, and then heated to 70 °C. After 2 h, clusters with strong orange-red fluorescence were formed.

Native polyacrylamide gel electrophoresis (PAGE) was carried out using discontinuous gels (1.5 mm × 80 mm × 70 mm). Resolving and stacking gels were prepared from 30 and 4 wt% acrylamide monomers. 200 μL of raw products was mixed with 20 μL of 5 vol% glycerol and then electrophoresis was run for 2 h at a constant voltage of 150 V at 4 °C. After electrophoresis, the band was cut from the gels and soaked in ultrapure water overnight at 4 °C to obtain the sample solution.

### 3.4. Peroxidase-like Activity

Quantities of 135 μL of the prepared clusters, 300 μL of 5 mM TMB and 200 μL of 100 mM H_2_O_2_ were added into 1.365 mL water. The mixture solution was then incubated in a water bath at 37 °C for 60 min before cooling to room temperature. The absorption spectra were recorded using a UV-Vis spectrophotometer.

### 3.5. Fluorescence Quenching Experiments with Adding H_2_O_2_ into the Cluster-TMB System

Quantities of 135 μL of the prepared clusters, 300 μL of 5 mM TMB and 200 μL of different concentrations of H_2_O_2_ were added to 1.365 mL water. The mixture solution was then incubated in a water bath at 37 °C for 60 min before cooling to room temperature. The emission spectra were recorded using a fluorescence analyzer.

### 3.6. Concentration-Dependent Peroxidase-like Activity of the Prepared Clusters

The concentration-dependent peroxidase-like activity of the gold clusters assays was measured at 37 °C with various concentrations of PGN (25, 50, 75, and 100 μM) in the presence of 100 mM H_2_O_2_ and 5 mM TMB at pH 3.0. The reaction was performed in a thermostatic mixer, and the absorbance of the samples at 652 nm was recorded via UV-Vis at different time points.

The pH of the solution containing 135 μL of clusters, 200 μL of 100 mM H_2_O_2_ and 300 μL of 5 mM TMB was adjusted from 2 to 7, and the absorbance of each sample at 652nm after retention in a thermomixer at 37 °C for 1h was recorded using a UV-Vis spectrophotometer.

The pH of the solution containing 135 μL of cluster solution, 200 μL of 100 mM H_2_O_2_ and 300 μL of 5 mM TMB was adjusted to 3, and the reaction was kept at different temperatures from 25 to 50 °C for 1 h. Then the absorbance of each sample at 652 nm was recorded using a UV-Vis spectrophotometer.

### 3.7. Determination of the Hydroxyl (·OH) Radical

Briefly, 135 μL of cluster solution was added to 2 mL water at pH 4. In this solution, 200 μL of 100 mM H_2_O_2_ and 800 μL of 6.25 mM terephthalic acid were mixed and incubated at 37 °C for 30 min. Then, the fluorescence spectra of the resultant solutions were measured with the excitation wavelength at 315 nm.

### 3.8. Steady-State Kinetic Assay of the Peroxidase-like Activity of the Clusters

Kinetic measurements of peroxidase-like properties were carried out by monitoring the time-dependent absorbance of ox-TMB at 652 nm using a UV–Vis spectrophotometer. The reaction kinetic data were measured according to the changing substrate concentration of TMB and H_2_O_2_. For the kinetic data relating TMB, 135 µL of cluster solution was added to a solution containing 200 µL of H_2_O_2_ (100mM); and 300 µL of different concentrations of TMB. Similarly, for the kinetic data relating H_2_O_2_, 135 µL of cluster solution was added to a solution containing 300 µL of TMB (1 mM); and 200 µL of different concentrations of H_2_O_2_.

The kinetic parameters (V_max_ and K_m_) for peroxidase-like activity of PGN were deduced from Michaelis-Menten equation:(1)$$v=\frac{V_{\max}\times \left[S\right]}{K_{m}+\left[S\right]}$$
where v, V_max_, [S] and K_m_ are the initial velocity, the maximal reaction velocity, the concentration of TMB and the Michaelis constant, respectively. K_cat_ was derived from K_cat_ = V_max_/[E], where [E] represents the concentration of the clusters (Au based).

### 3.9. Cytotoxicity Experiment

MTT (5-dimethylthiazol-2-yl-2,5-diphenyltetrazolium bromide) assays were performed. NIH 3T3 and HeLa cells were passed and plated to ca. 70% confluence in 96-well plates 24 h before treatment. Then, DMEM (Dulbecco’s Modified Eagle Medium) with 10% FBS (fetal bovine serum) was removed and replaced with fresh DMEM, and the cluster solution was then added to obtain final concentrations of 0, 20, 40, 60, 80 and 100 μM (Au-based). The treated cells were incubated for 12/24 h at 37 °C under 5% CO_2_. Then, the cells were treated with 5 mg/mL MTT (10 μL/well) and incubated for another 4 h (37 °C, 5% CO_2_). The cells were then dissolved in DMSO (150 μL/well), and the absorbance at 570 nm was recorded. Cell viability (%) was analyzed based on the following equation:Cell viability % = OD_570_ (sample)/OD_570_ (control) × 100%
where OD_570_ (sample) is the optical density of the wells treated with various concentration of clusters and OD_570_ (control) is that of the wells treated with DMEM containing 10% FBS (fetal bovine serum). The percentage of cell survival values were relative to untreated control cells. Each individual cytotoxicity experiment was repeated three times.

### 3.10. Cell Culture and Confocal Fluorescence Imaging

NIH 3T3 and HeLa cells were cultured in Dulbecco’s Modified Eagle Medium (DMEM) supplemented with 10% FBS (fetal bovine serum), penicillin (100 μg/mL), and streptomycin (100 μg/mL) at 37 °C in a humidified atmosphere of 5% CO_2_ and 95% air. Cytotoxicity assays showed that the clusters are biocompatible at low concentrations, so a concentration of 60 μM was chosen for fluorescence imaging in cells. The cells were incubated with 60 μM clusters for 2 h, then washed 3 times with PBS buffer. Then, cell imaging was recorded using a confocal microscope (TCS SP8 DIVE).

## 4. Conclusions

Nano-enzymes have recently attracted increasing research interest for their high catalytic activity and capacity to overcome the disadvantages of natural enzymes, whereas comprehensive synthetic procedures and harsh reaction conditions (such as high temperature, long reaction time,) significantly restrict the application of these materials. In this study, gold nanoclusters protected by oligopeptides (DGEAGC) were prepared using a mild one-pot synthesis method, i.e., via simply reacting the oligopeptide and HAuCl_4_ in aqueous solution at 70 °C for only two hours. The UV-Vis, PAGE separation, HRTEM, XPS, and FTIR analysis demonstrated the monodispersity of the prepared clusters. The prepared clusters showed strong orange-red emission and excellent chemical stability. PGN can oxidize TMB to produce blue ox-TMB in the presence of H_2_O_2_, indicating peroxidase activity, which conforms to the steady-state kinetic equation of the enzyme. In particular, the good biocompatibility and cell imaging properties indicate the potential of gold nanoclusters as a multifunctional platform.

## Data Availability

All data are included in the article and/or Appendix A.

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
