# Peer review of "An Oligopeptide-Protected Ultrasmall Gold Nanocluster with Peroxidase-Mimicking and Cellular-Imaging Capacities"

_molecules, 2022, doi:10.3390/molecules28010070_

Round 1

Reviewer 2 Report

The presented manuscript is interesting and on the top topic of the nanocluster-based researcher. During the review of the article, I found some comments and questions:

1) I recommend to re-write the whole Abstract. It is rather general...

2) Several article is missing from the Introduction about the enzyme-like activity of the noble metal nanoclusters. I recommend to use the following references:

doi: 10.1002/bio.3472

doi: 10.1007/s00216-019-01844-9

doi: 10.1007/s00604-019-3395-8

doi:10.3389/fchem.2020.00219

3) In lines 78-79 the following was written: "The average fluorescence lifetime of 6.14 μs (Figure S3), longer than the nanosecond scale luminescence lifetime for most ultrasmall metal nanoclusters." 

this is not surprising because it is well-known that the peptide-stabilized clusters show μs-ranged lifetimes, which can be easily tuned by the peptide sequences. I recommend the re-edition of this sentence.

4) I recommend to present the kinetic curves (de spectra series during the kinetic measurements) in the main text not only the calculated data.

5) The purity of the chemicals is missing. It should be complete.

6) The conclusion is too short. I suggest to complete this section.

After these minor revisions, I recommend to publish.

Reviewer 3 Report

The authors report the synthesis, characterization and catalytic activity of a peptide-coated gold nanocluster that displays catalase activity. The characterization is thorough and the work is publishable with a few minor revisions/additions:    

Major english revisions required

IR spectral interpretation: The conclusion that cleavage of the SH bond is not warranted. Cleavage of the SH bond is not required for diminishing IR peak as binding to gold through the sulfur lone pair electrons can cause the same effect. Thiols can bind gold without losing the proton.    The authors should discuss the pH dependence of the mano-cluster fluorescence in terms of ionizable groups on the peptide and suggest a reason for the dependence.   How do the authors know that glutathione is not binding to the gold cluster through its thiol and restoring the nano-cluster fluorescence? Glutathione might indeed be reacting with TMB, but could also be interacting with the gold cluster.   Why is 37C chosen? Figure 3B doesn’t show an obvious reason for the choice as the data increases linearly

Round 2

Reviewer 1 Report

The author's are improved/enhanced their revised manuscript, The author's are answered my concern comments and suggestions 

The current form of the manuscript can be suitable to publish, i recomendanded to the editor.